# Bridging the gap between presynaptic hair cell function and neural sound encoding

**Lina María Jaime Tobón**[1,2,3,4]*[†], **Tobias Moser**[1,2,3,4]*

[1]Auditory Neuroscience and Synaptic Nanophysiology Group, Max Planck Institute for Multidisciplinary Sciences, Göttingen, Germany; [2]Institute for Auditory Neuroscience, University Medical Center Göttingen, Göttingen, Germany; [3]Collaborative Research Center, University of Göttingen, Göttingen, Germany; [4]Cluster of Excellence 'Multiscale Bioimaging of Excitable Cells', Göttingen, Germany

*For correspondence:
ljaimet@mpinat.mpg.de (LMJT);
tmoser@gwdg.de (TM)

Present address: [†]Institute for Neurosciences Montpellier, Institut National de la Santé et de la Recherche Médical, University of Montpellier, Montpellier, France

Competing interest: The authors declare that no competing interests exist.

## eLife Assessment

This **fundamental** study advances substantially our understanding of sound encoding at synapses between single inner hair cells of the mouse cochlea and spiral ganglion neurons. Dual patch-clamp recordings-a technical tour-de force-and careful data analysis provide **compelling** evidence that the functional heterogeneity of these synapses contributes to the diversity of spontaneous and sound-evoked firing by the neurons. The work will be of broad interest to scientists in the field of auditory neuroscience.

**Abstract** Neural diversity can expand the encoding capacity of a circuitry. A striking example of diverse structure and function is presented by the afferent synapses between inner hair cells (IHCs) and spiral ganglion neurons (SGNs) in the cochlea. Presynaptic active zones at the pillar IHC side activate at lower IHC potentials than those of the modiolar side that have more presynaptic $Ca^{2+}$ channels. The postsynaptic SGNs differ in their spontaneous firing rates, sound thresholds, and operating ranges. While a causal relationship between synaptic heterogeneity and neural response diversity seems likely, experimental evidence linking synaptic and SGN physiology has remained difficult to obtain. Here, we aimed at bridging this gap by ex vivo paired recordings of murine IHCs and postsynaptic SGN boutons with stimuli and conditions aimed to mimic those of in vivo SGN characterization. Synapses with high spontaneous rate of release (*SR*) were found predominantly on the pillar side of the IHC. These high *SR* synapses had larger and more temporally compact spontaneous EPSCs, lower voltage thresholds, tighter coupling of $Ca^{2+}$ channels and vesicular release sites, shorter response latencies, and higher initial release rates. This study indicates that synaptic heterogeneity in IHCs directly contributes to the diversity of spontaneous and sound-evoked firing of SGNs.

## Introduction

Chemical synapses represent diverse and plastic neural contacts that are adapted to the specific needs of neural computation. Synaptic diversity is expressed across the nervous system, within a given circuit and even within the same neuron (recent reviews in *Nusser, 2018*; *Wichmann and Kuner, 2022*). Synaptic diversity occurs at various levels: from synapse shape and size, to the ultrastructure of pre- and postsynaptic specializations, to their molecular composition. The auditory system harbors striking examples of synaptic diversity. Glutamatergic ribbon synapses in the cochlea, calyceal

**eLife digest** From the rustling of falling leaves all the way to a roaring jet engine, our sense of hearing allows us to recognise a wide range of sounds that vary in pitch and intensity. Two groups of inner ear cells, known as the inner hair cells and the spiral ganglion neurons, perform this feat by encoding sounds into signals that can be processed by the nervous system.

This mechanism relies on inner hair cells detecting sound vibrations and then causing spiral ganglion neurons to 'fire' electrical signals that can be relayed to the brain. Each inner hair cell connects to multiple spiral ganglion neurons through contact points called synapses, where information corresponding to specific sounds is transmitted. For any given pitch, different groups of spiral ganglion neurons encode different sound intensities (that is, loud versus soft sounds). Whether this is because these groups are fundamentally different in some way or because the synapses between neurons and hair cells have different properties remained to be elucidated.

To investigate this question, Jaime Tobón and Moser examined the mechanisms underpinning sound encoding in the inner ear of mice, using tissue preparations containing inner hair cells and spiral ganglion neurons with fully intact synapses.

Measuring the electrical properties on both the inner hair cell and spiral ganglion neuron side of individual synapses revealed differences in the levels of spontaneous activity of the synapses. Synapses with higher spontaneous activity detected softer stimuli, whereas those with lower rates responded only to stronger stimulation.

Each type of synapse formed at different locations on the surface of inner hair cells, and they had different electrical properties that mirrored the firing diversity of the spiral ganglion neurons. In other words, it is the inner hair cell 'side' of the synapses that dictates the different responses of the neurons connected to them. To diversify the response of their synapses, the inner hair cells relied on variations in synaptic properties (such as voltage-dependent activation thresholds and the coupling of calcium channels and vesicular release sites) that determine how sensitive a cell is to an electric signal, and how quickly and efficiently it can react to it.

These results shed new light on the biological mechanism of sound encoding, a process fundamental to our sense of hearing. In the future, Jaime Tobón and Moser hope that this knowledge may eventually inform the development of better aids and treatments for hearing loss patients.

synapses in the brainstem, and bouton synapses throughout the central auditory system differ greatly from each other (*Moser et al., 2006*; *Wichmann and Kuner, 2022*). Beyond the diversity across synapses formed by different neurons and at different places (e.g. different regions of the brain or frequency [tonotopic] places of the cochlea; *Johnson et al., 2017*), heterogeneity exists even among auditory synapses formed by an individual presynaptic inner hair cell (IHC) with its 5–30 postsynaptic spiral ganglion neurons (SGNs, reviews in *Gómez-Casati and Goutman, 2021*; *Meyer and Moser, 2010*; *Moser, 2020*; *Reijntjes and Pyott, 2016*). Synaptic heterogeneity has been found at different tonotopic positions of the cochlea and is a candidate mechanism for how the cochlea decomposes acoustic information (*Grant et al., 2010*; *Meyer et al., 2009*; *Ohn et al., 2016*; *Özçete and Moser, 2021*). For example, the cochlea might use heterogeneous afferent synapses to break down sound intensity information into complementary spike rate codes of SGNs that have been reported along the tonotopic axis of the cochlea for several species (*Huet et al., 2016*; *Kiang et al., 1965*; *Sachs and Abbas, 1974*; *Taberner and Liberman, 2005*; *Winter et al., 1990*).

Decades of in vivo recordings from single SGNs have demonstrated functional diversity of SGNs with comparable frequency tuning, i.e., receiving input from IHCs at a given tonotopic place or potentially even the same IHC. Such functional diversity is present in both spontaneous and sound-evoked firing. The spontaneous firing rate (SR) in the absence of sound varies from less than 1 spikes/s to more than 100 spikes/s (*Barbary, 1991*; *Evans, 1972*; *Kiang et al., 1965*; *Schmiedt, 1989*; *Taberner and Liberman, 2005*). In response to increasing sound pressure, SGNs with high SR show a low sound threshold and a steep rise in the spike rate to increasing sound intensities until the rate saturates. SGNs with low SR have a higher sound thresholds, shallower spike rate rise, and saturate at higher sound intensities (*Ohlemiller et al., 1991*; *Winter et al., 1990*). Additionally, SGNs show differences in their excitability (*Crozier and Davis, 2014*; *Markowitz and Kalluri, 2020*; *Smith et al., 2015*),

morphological features (*Liberman, 1980*; *Merchan-Perez and Liberman, 1996*; *Tsuji and Liberman, 1997*) and heterogeneous molecular profiles (*Li et al., 2020*; *Petitpré et al., 2020*; *Petitpré et al., 2018*; *Shrestha et al., 2018*; *Sun et al., 2018*). Yet, it has remained challenging to demonstrate a causal link of a candidate mechanism to the physiological SGN diversity.

One common approach has been to capitalize on a pioneering study that employed in vivo labeling of physiologically characterized SGNs in cats and proposed that synapses formed by low and high SR SGNs segregate on the basal IHC pole (*Liberman, 1982*). High SR SGNs preferentially contacted the pillar side of the IHC (facing pillar cells), while low SR SGNs synapsed on the opposite, modiolar side of the IHC (facing the cochlear modiolus). Interestingly, a segregation has also been found for afferent and efferent synaptic properties, as well as molecular and biophysical SGN properties (*Frank et al., 2009*; *Grant et al., 2010*; *Hua et al., 2021*; *Kantardzhieva et al., 2013*; *Liberman et al., 2011*; *Markowitz and Kalluri, 2020*; *Merchan-Perez and Liberman, 1996*; *Meyer et al., 2009*; *Neef et al., 2018*; *Ohn et al., 2016*; *Özçete and Moser, 2021*; *Shrestha et al., 2018*; *Sun et al., 2018*; *Yin et al., 2014*). For instance, type $I_b$ and $I_c$ SGNs preferentially synapse on the modiolar side (*Sherrill et al., 2019*; *Shrestha et al., 2018*; *Sun et al., 2018*) and show low SR (*Siebald et al., 2023*). Pillar synapses are preferentially formed by type $I_a$ SGNs (*Shrestha et al., 2018*; *Siebald et al., 2023*), have smaller IHC active zones (AZs) (*Liberman et al., 2011*; *Ohn et al., 2016*; *Reijntjes et al., 2020*), and activate at voltages as low as the IHC resting potential (*Ohn et al., 2016*; *Özçete and Moser, 2021*). The low voltage of activation of release at pillar synapses shown ex vivo could underly the high SR and low sound threshold of firing of SGNs found in vivo but direct demonstration of such a link has been missing.

Here, we aimed to bridge the gap between ex vivo presynaptic physiology and in vivo SGN neurophysiology. We performed paired IHC and SGN bouton recordings in acutely explanted organs of Corti from hearing mice with stimuli and conditions aimed to mimic in vivo SGN

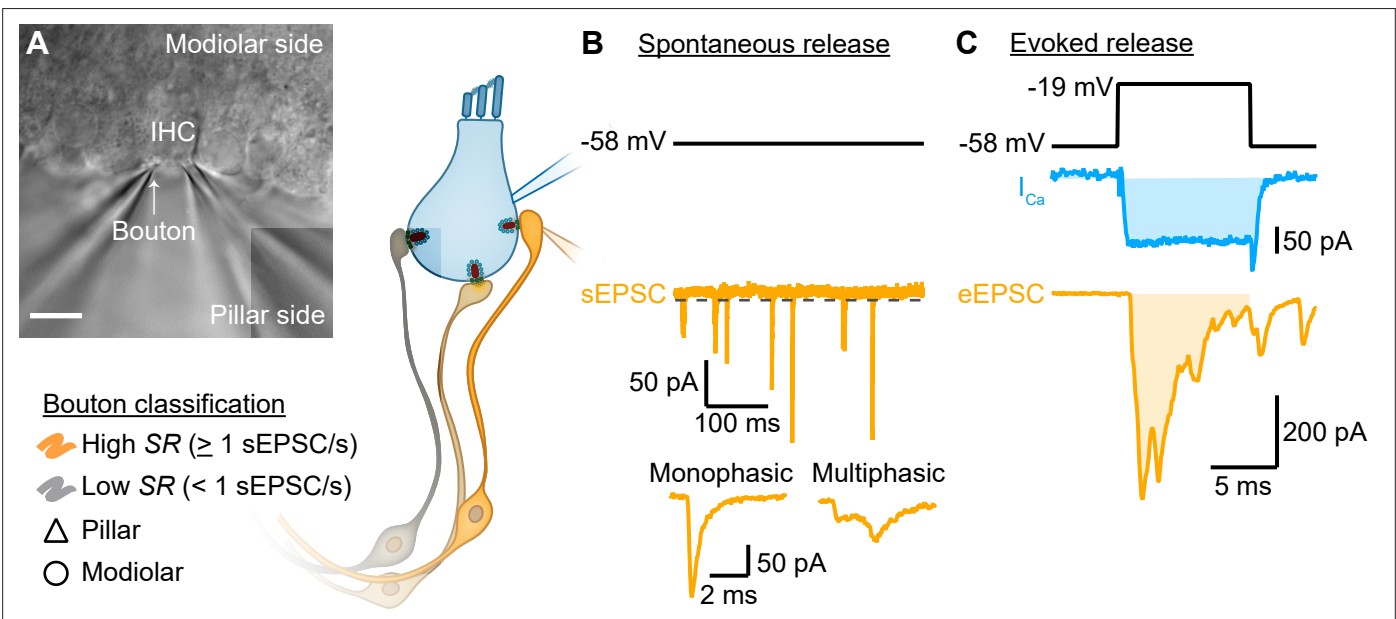

**Figure 1.** Paired inner hair cell (IHC)- spiral ganglion neuron bouton patch-clamp recordings to study the release properties of individual IHC ribbon synapses as a function of synapse position. (**A**) Differential interference contrast (DIC) image of an explanted murine organ of Corti. In this example, supporting cells from the pillar side were removed to gain access to the IHCs and their contacting boutons. The recorded boutons were classified based on their position (△ pillar or ◯ modiolar) and on their spontaneous rate (*SR*) (Low *SR*<1 sEPSC/s vs High *SR*≥1 sEPSC/s). Scale bar: 10 μm. (**B**) Spontaneous release was recorded in absence of stimulation (i.e. IHC holding potential = –58 mV; *Supplementary file 1*; dashed line represents the threshold for sEPSC detection). sEPSCs were classified as monophasic (a steady rise to peak and monoexponential decay, temporally more compact) or as multiphasic (multiple inflections and slowed raising and decaying kinetics, non-compact). (**C**) Evoked release: depolarizing pulses (black trace) were used to trigger whole IHC $Ca^{2+}$ influx ($I_{Ca}$, blue trace) and ensuing release of neurotransmitter that evoked EPSCs (eEPSCs, light orange trace). $Ca^{2+}$ charge and eEPSC charge were estimated by taking the integral of the currents (shaded light blue and light orange areas).

The online version of this article includes the following figure supplement(s) for figure 1:

**Figure supplement 1.** Passive electrical properties of inner hair cells (IHCs) and spiral ganglion neuron (SGN) boutons.

characterization. This approach tightly controls IHC Ca$^{2+}$ influx and records the postsynaptic SGN response to glutamate release at a single afferent synapse. Using the rate of spontaneous excitatory postsynaptic currents (rate of sEPSCs, *SR*) as a surrogate for SGN SR, we demonstrate that high *SR* synapses have larger and more compact sEPSCs as well as lower voltage thresholds, shorter latencies of evoked EPSCs (eEPSCs), tighter Ca$^{2+}$ channel coupling to vesicle release, and higher initial release rates. 90% of these high *SR* synapses were located on the pillar side of the IHC. Our findings suggest that synaptic heterogeneity accounts for much of the SGN firing diversity at a given tonotopic position.

## Results

Simultaneous paired patch-clamp recordings from IHCs and one of the postsynaptic SGN boutons were performed on mice after the onset of hearing (postnatal days [p] 14–20). We performed perforated-patch whole-cell configuration from IHCs, held at their presumed physiological resting potential (–58 mV; *Johnson, 2015*), and ruptured-patch whole-cell recordings from one of the postsynaptic SGN boutons (*Figure 1*). Due to the technical difficulty of establishing the paired recording, typically only one bouton was recorded per IHC. Recordings were made at body temperature and in artificial perilymph-like solution (*Wangemann and Schacht, 1996*). To establish the paired recording, we approached boutons facing either the pillar or the modiolar side of the IHC in an effort to elucidate synaptic differences between both sides (*Figure 1A*). We nickname the synapse location as 'pillar' and 'modiolar' based on the DIC image, but note that efforts to stain and image the recorded boutons by fluorescence microscopy were not routinely successful. In addition, the recorded boutons were classified based on their spontaneous rate of synaptic transmission (*Figure 1B*, *Figure 2*, and related figure supplements; *SR,* Low *SR*<1 sEPSC/s vs High *SR*≥1 sEPSC/s according to *Taberner and Liberman, 2005*). We then performed an *in-depth* biophysical analysis of evoked release (*Figure 1C*, *Figure 3*, *Figure 4*, and related figure supplements).

### Spontaneous synaptic transmission

In order to recapitulate synaptic transmission in the absence of sound stimulation, we held the IHC at their presumed physiological resting potential (–58 mV; *Johnson, 2015*). Since Ca$_V$1.3 Ca$^{2+}$ channels activate at low voltages (–65 to –45 mV; *Koschak et al., 2001*; ; *Picher et al., 2017a*; *Platzer et al., 2000*; *Xu and Lipscombe, 2001*), their open probability at the IHC resting potential is thought to be sufficient to trigger spontaneous release (*Glowatzki and Fuchs, 2002*; *Özçete and Moser, 2021*; *Robertson and Paki, 2002*). Under our experimental conditions, spontaneous, i.e., excitatory postsynaptic currents in the absence of IHC stimulation (sEPSCs) were observed in 23 of 33 pairs (*Figure 1B*, *Figure 2*, 20 recordings were targeted to the pillar side and 13 to the modiolar side). We used the rate of sEPSCs (*SR*) as a surrogate of SGN SR as sEPSCs trigger SGN firing with >90% probability (*Rutherford et al., 2012*). Regardless of all efforts to maintain the physiological integrity in the ex vivo experiments, we expect our estimated rates of sEPSC to underestimate the SGN SR for the same age group (*Wong et al., 2013*). For each paired recording, we quantified the *SR* during 5 or 10 s of a continuous recording, or during the segment before and after the step depolarization protocols (*Supplementary file 1*). Amplitudes of sEPSCs typically varied from around –10 to –400 pA across all recorded IHC-SGN synapses (*Figure 2B*). The amplitude histogram for all pairs was slightly skewed toward smaller amplitudes (skewness of 1.06) with a coefficient of variation (CV) of 0.68 (a similar distribution was obtained from 12 bouton-only recordings during which the IHC was not patched [*Figure 2—figure supplement 1A*]). The charge distribution for all pairs displayed a prominent peak at 40 fC, with a skewness of 2.00 and a CV of 0.77. *SR* ranged from 0 to about 18 sEPSC/s (*Figure 2H*; a similar range of 0 to about 16 sEPSC/s was recorded without patch-clamping the IHC). The *SR* distribution was highly skewed, with a median of 0.2 sEPSC/s. Following the spontaneous firing rate classification of mouse SGNs by *Taberner and Liberman, 2005*, we classified the synapses into low (<1 sEPSC/s, ~70%) or high (≥1 sEPSC/s, ~30%) *SR* synapses (see *Figure 2A* for examples). Next, we analyzed the recordings as a function of (i) synapse position and (ii) rate of spontaneous synaptic transmission.

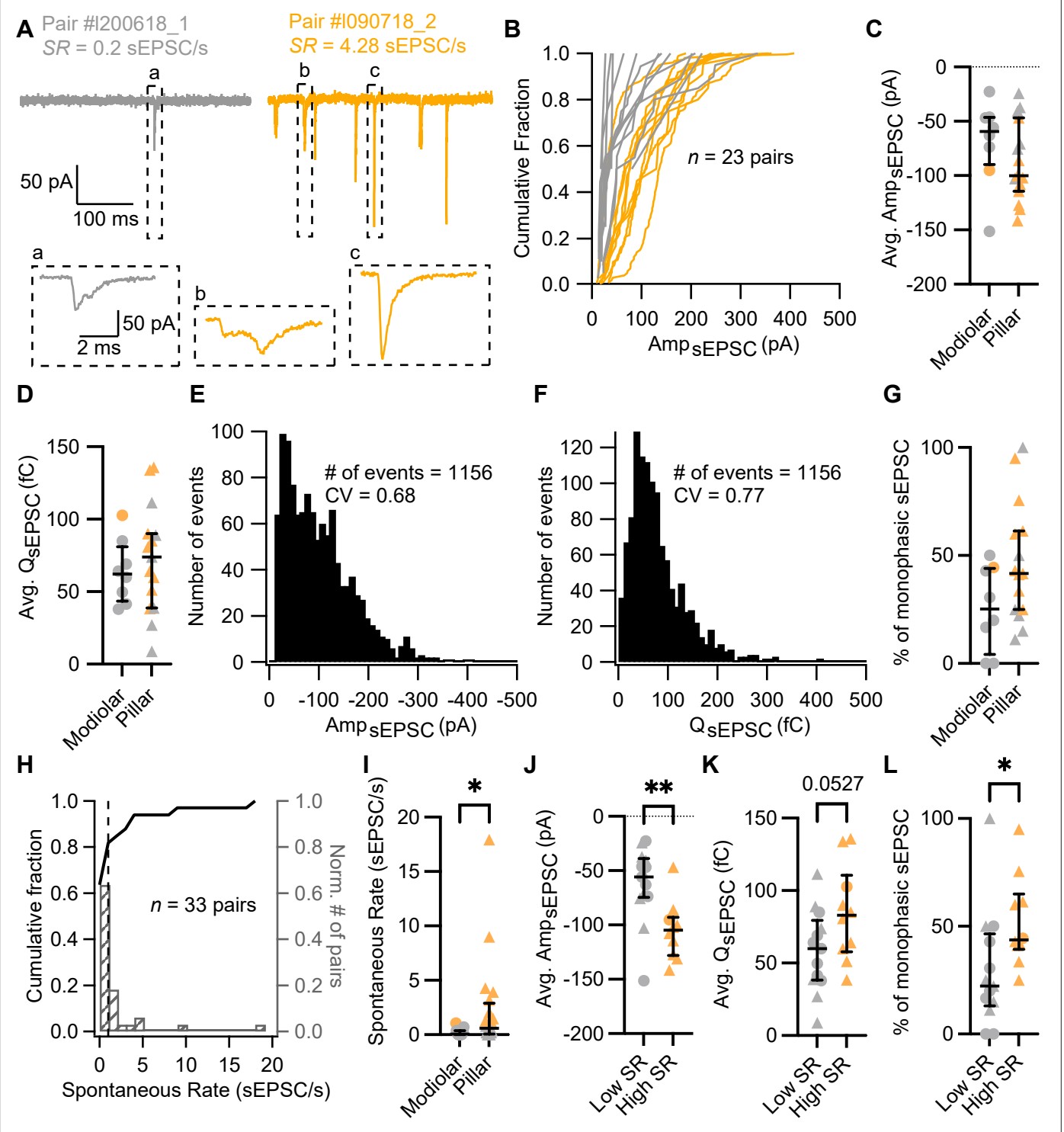

**Figure 2.** Synapses with high spontaneous release have larger and more monophasic spontaneous EPSCs (sEPSCs). (**A**) sEPSCs recorded in the absence of stimulation (i.e. inner hair cell [IHC] holding potential = –58 mV) from two exemplary paired recordings with different spontaneous rate (*SR*: gray for low *SR*, orange for high *SR*). 'Pair #' identifies individual paired recordings. Insets show the selected sEPSCs in an expanded time scale. (**a, b**) correspond to multiphasic sEPSCs, while (**c**) represents a typical monophasic sEPSC. (**B**) Cumulative sEPSC amplitude plots for 23 paired synapses that had spontaneous release. (**C–D**) Average sEPSC amplitude (**C**) and charge (**D**) from individual synapses recorded from the pillar or modiolar side of the IHC. (**E–F**) Pooled sEPSC amplitude (**E**) and charge (**F**) distributions show a distinct peak at –40 pA and 40 pC, respectively. Bin size: 10 pA or pC. (**G**) Percentage of monophasic sEPSCs in pillar and modiolar synapses. (**H**) Cumulative fraction (left axis) and normalized histogram (right axis) of the

*Figure 2 continued on next page*

*Figure 2 continued*

spontaneous rate (bin size is 1 sEPSC/s) of 33 pairs. (**I**) Pillar synapses had higher rates of sEPSCs. (**J–L**) High *SR* synapses had significantly larger sEPSC amplitudes (**J**), a tendency to bigger sEPSC charges (**K**) and higher percentages of monophasic sEPSCs (**L**). Panels **G, I–L** show individual data points with the median and interquartile range overlaid (line). Synapses were classified as △ pillar or ○ modiolar, and as Low *SR*<1 sEPSC/s≤High *SR*.

The online version of this article includes the following figure supplement(s) for figure 2:

**Figure supplement 1.** Pillar and high spontaneous release synapses have sEPSCs with faster rising times.

## Position dependence of synaptic transmission

Of the 33 obtained paired recordings, 20 were classified as pillar synapses. The mean amplitude (*Figure 2C*) and charge (*Figure 2D*) of the sEPSCs were comparable between modiolar and pillar SGN boutons (Amp$_{sEPSC}$ –69.37±13.91 pA [n=8 modiolar] vs –87.21±9.49 pA [n=15 pillar]; p=0.2913, unpaired t-test; $Q_{sEPSC}$ 63.74±7.77 fC [modiolar] vs 72.54±9.57 fC [pillar]; p=0.5463, unpaired t-test). sEPSC of pillar SGN boutons showed significantly shorter 10–90% rise times than the modiolar ones (*Figure 2—figure supplement 1B and C*; 0.38±0.04 ms vs 0.57±0.06 ms; p=0.0111, unpaired t-test), yet similar decay times and full-width half-maximum (FWHM, *Figure 2—figure supplement 1B, D–E*; p=0.7997 and p=0.9198, respectively, unpaired t-test). As a second approach to sEPSC properties, we quantified the percentage of monophasic (or temporally more compact) sEPSCs (*Chapochnikov et al., 2014*; *Glowatzki and Fuchs, 2002*) and found a non-significant trend toward higher percentages of monophasic sEPSCs for pillar synapses (*Figure 2G*; 25.57 ± 6.9% for modiolar vs 46.65 ± 7.06% for pillar; p=0.0681, unpaired t-test). Finally, *SR* was significantly higher for the pillar synapses (*Figure 2I*, mean *SR* of 2.30±0.96 sEPSC/s; median 0.59; n=20) compared to modiolar ones (*Figure 2I*, mean *SR* of 0.22±0.09 sEPSC/s; median 0.05; n=13; p=0.0311, Mann-Whitney U test).

## Relation of synaptic properties to the rate of spontaneous synaptic transmission

High *SR* synapses had significantly larger sEPSCs (*Figure 2J*, *Figure 2—figure supplement 1Fi*; average sEPSC amplitude of –105.2±8.47 pA for high *SR* [n=10] vs –62.39±9.67 pA for low *SR* [n=13]; p=0.0042, unpaired t-test). sEPSC charge tended to be larger in high *SR* SGN synapses (*Figure 2K*, *Figure 2—figure supplement 1Fii*; $Q_{sEPSC}$ 84.23±10.38 fC for high *SR* vs 58.14±7.79 fC for low *SR*; p=0.0527, unpaired t-test). The fraction of monophasic sEPSCs was significantly higher in high *SR* synapses (*Figure 2L*; 52.08 ± 6.65%; median 43.71%) compared to low *SR* synapses (29.49 ± 7.43%; median 22.22%; p=0.0185, Mann-Whitney U test). High *SR* synapses also showed a significantly faster 10–90% rise times (*Figure 2—figure supplement 1G*; 0.36±0.04 ms) than low *SR* synapses (0.51±0.05 ms; p=0.0420, unpaired t-test).

Other sEPSCs kinetics, such as decay time constant and FWHM, were not different between low and high *SR* pairs (*Figure 2—figure supplement 1H and I*; p=0.7969 and p=0.9948, respectively, unpaired t-test). Taken together, these results indicate that high *SR* synapses are characterized by sEPSCs with larger amplitudes, faster rising times, and a more compact waveform, while significant differences of pillar and modiolar synapses were limited to sEPSC rise times. Yet, 9 out of 10 synapses with *SR*≥1 sEPSC were located on the pillar side of the IHC.

## Evoked synaptic transmission differs between afferent synapses with high and low *SR*

Next, we compared the physiology of afferent synapses with high and low *SR* by adapting stimulation protocols routinely employed for in vivo characterization of sound encoding by SGNs. We used step depolarizations to emulate physiological receptor potentials given that mature IHCs of the 'high-frequency' mouse cochlea have graded receptor potentials that primarily represent the rectified envelope of an acoustic stimulus (i.e. the DC component; *Russell and Sellick, 1978*).

### Stimulus intensity encoding at IHC synapses

Sound intensity encoding by SGNs primarily relies on a spike rate code: the average discharge rate increases with the strength of the acoustic stimuli from threshold to saturation of the response. These so-called 'rate level functions' are typically analyzed by fitting a sigmoidal function, of which the range

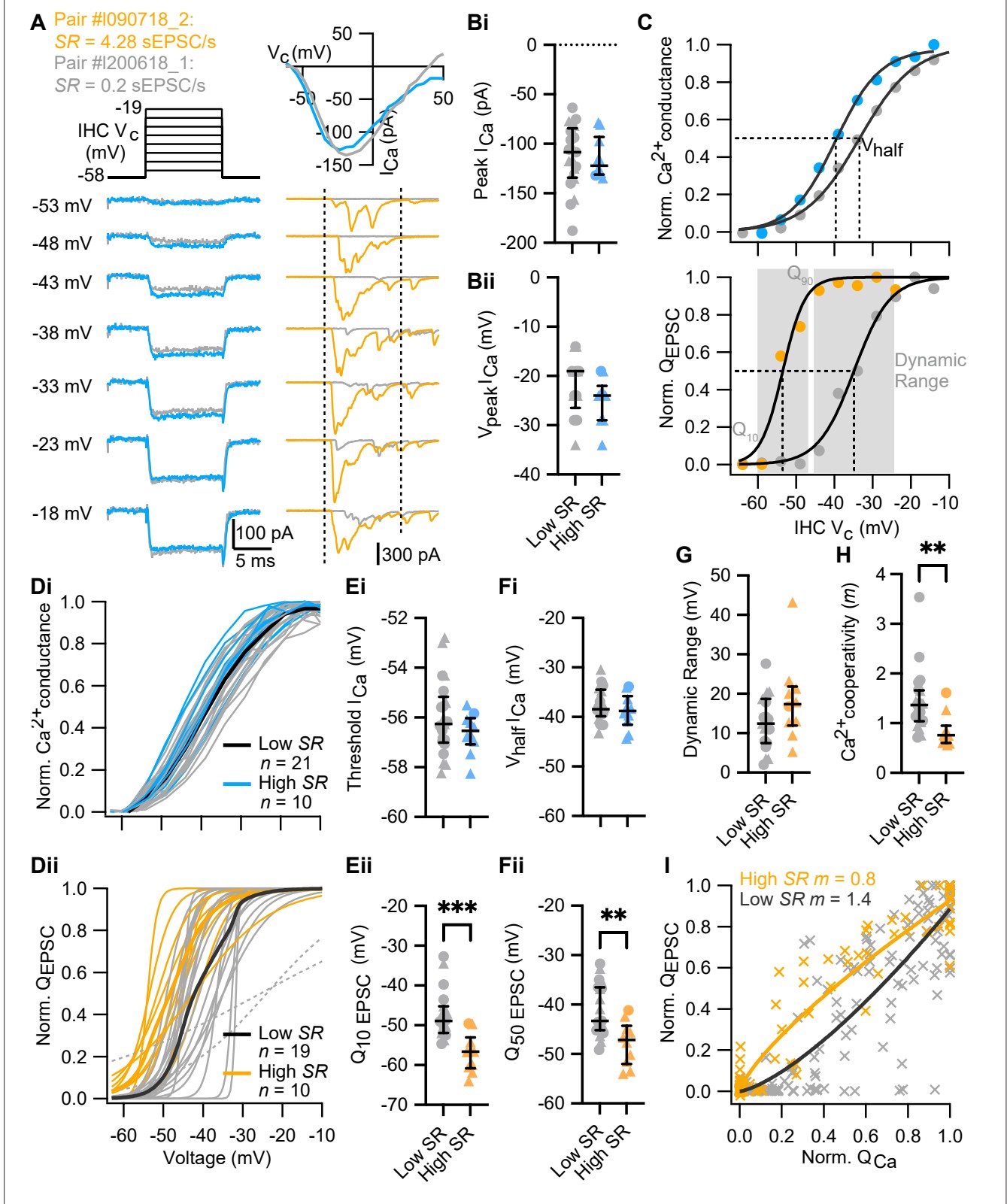

**Figure 3.** High spontaneous release synapses activate at lower voltages and show tighter $Ca^{2+}$ channel coupling of synaptic release. (**A**) Voltage-protocol (top left), inner hair cell (IHC) $Ca^{2+}$ current ($I_{Ca}$, bottom left and top right, blue and gray) and evoked EPSCs (eEPSCs) (bottom right, orange and gray) of a high and a low *SR* pair, respectively, in response to 10 ms depolarizations (dashed vertical lines on top of eEPSC data indicate the onset and offset of the depolarization) to different potentials ranging from –58 to –19 mV in 5 mV steps. The upper right panel shows the current-

*Figure 3 continued*

voltage relationships for the two pairs. (**Bi–Bii**) The peak of whole-cell $Ca^{2+}$ current (**Bi**) and the voltage eliciting maximum $Ca^{2+}$ current (**Bii**) of IHCs were comparable between high and low *SR* synapses. (**C**) Upper panel: Fractional activation of the $Ca^{2+}$ channels (blue and gray data points from the examples shown in A) was obtained from the normalized chord conductance. Voltage of half-maximal activation ($V_{half}\ I_{Ca}$; dotted line) and voltage sensitivity of activation (slope) were determined using a Boltzmann fit (black trace) to the activation curve. Lower panel: Release-intensity curve (orange and gray data points from the examples shown in A) was obtained from the $Q_{EPSC}$ for each depolarization step. A sigmoidal function (black trace) was fitted to obtain the voltage of half-maximal synaptic release ($Q_{50\ EPSC}$; dotted line) and the voltage sensitivity of the release (slope), as well as the dynamic range for which the exocytosis changes from 10% to 90% (gray area). (**Di–Dii**) Voltage dependence of whole-cell $Ca^{2+}$ channel activation (activation curve; **Di**) and fits to release-intensity curves (**Dii**) for 31 synapses. Averages (thick lines) and individual curves (thin lines) are overlaid. The release-intensity curve of two low *SR* pairs could not be fitted (gray dotted lines). (**Ei–Fi**) The threshold of $Ca^{2+}$ influx (**Ei**) and $V_{half}\ I_{Ca}$ (**Fi**) did not differ between low and high *SR* synapses. (**Eii–Fii**) Voltage of 10% maximum release ($Q_{10\ EPSC}$, **Eii**) and $Q_{50\ EPSC}$ (**Fii**) were significantly more hyperpolarized in high *SR* synapses. (**G**) Dynamic range of release was comparable between low and high *SR* synapses. (**H**) $Ca^{2+}$ cooperativity (m) estimated from fitting a power function to the $Q_{EPSC} – Q_{Ca}$ relationship for each individual synapse (see ***Figure 3—figure supplement 2***) was significantly lower in high *SR* synapses. (**I**) Scatter plot of normalized $Q_{EPSC}$ vs the corresponding normalized $Q_{Ca}$. The solid lines are a least-squares fit of a power function ($Q_{EPSC} = a(Q_{Ca})^m$) to the data, yielding $m_{high\ SR}$ of 0.8 and $m_{low\ SR}$ of 1.4. Panels B, E–H show individual data points with the median and interquartile range overlaid (line). Synapses were classified as △ pillar or ○ modiolar, and as Low *SR*<1 sEPSC/s≤High *SR*.

The online version of this article includes the following figure supplement(s) for figure 3:

**Figure supplement 1.** The voltage dependence of synaptic release does not differ significantly between modiolar and pillar synapses, but the $Ca^{2+}$ dependence does.

**Figure supplement 2.** Apparent $Ca^{2+}$ dependence of neurotransmitter release at individual synapses in the range of inner hair cell (IHC) receptor potentials.

of sound pressure level between 10% and 90% of the maximal discharge rate represents the operational or dynamic range (***Sachs and Abbas, 1974***; ***Taberner and Liberman, 2005***; ***Winter et al., 1990***). To understand stimulus intensity coding at mouse IHC synapses, we measured whole-cell IHC $Ca^{2+}$ currents and the eEPSCs of SGNs in 31 paired recordings. We stimulated the IHC with 10 ms depolarizations to different potentials ranging up to 57 mV in 5 mV steps (IV protocol; ***Figure 3A***). We deemed it incompatible with a reasonable productivity of the technically challenging, low-throughput paired recordings to combine them with imaging of $Ca^{2+}$ at single AZs. Therefore, this study relies on analysis of the presynaptic $Ca^{2+}$ influx at the level of the whole IHC (i.e. summing over all synapses and a low density of extrasynaptic $Ca^{2+}$ channels, ***Frank et al., 2009***; ***Wong et al., 2014***). IHCs with synapses classified as high (n=10) or low *SR* (n=21) had similar $Ca^{2+}$ current-voltage (IV) relationships: comparable maximal $Ca^{2+}$ currents (***Figure 3Bi***; p=0.6939, Mann-Whitney U test) elicited at similar potentials (***Figure 3Bii***; p=0.1795, unpaired t-test) and comparable reversal potentials (***Figure 3—figure supplement 1A***; p=0.4034, unpaired t-test). The fractional activation of $Ca^{2+}$ channels was determined from the normalized chord conductance of the IHC. Fitting a Boltzmann function to these activation curves (***Figure 3C***, upper panel), we obtained the voltages of half-maximal activation ($V_{half}\ I_{Ca}$) and the voltage-sensitivity of activation (slope) of the IHC $Ca^{2+}$ channels.

'Release-stimulus intensity' curves, akin of an ex vivo representation of the SGN rate-level function, were constructed from the normalized $Q_{EPSC}$ response obtained during the IV protocol (***Figure 3—figure supplement 1C***). The voltage dependence of synaptic vesicle (SV) release per AZ was approximated by the fit of a sigmoidal function to the individual release-intensity curves (***Figure 3C***, lower panel and ***Figure 3Dii***). From these sigmoidal fits, we obtained voltage of 10%-maximal release ($Q_{10\ EPSC}$), voltage of half-maximal release ($Q_{50\ EPSC}$), voltage of 90%-maximal release ($Q_{90\ EPSC}$), and the voltage sensitivity of release (slope). For two low *SR* paired recordings (***Figure 3Dii***, gray dotted lines), a sigmoidal function did not properly fit the release-intensity curve (assessed by visual inspection) which led us to exclude them from the statistical analysis.

The voltage dependence of activation of whole-cell $Ca^{2+}$ influx was similar between IHCs contacted by high and low *SR* boutons: threshold of $Ca^{2+}$ influx (***Figure 3Ei***; p=0.2393, unpaired t-test), $V_{half}\ I_{Ca}$ (***Figure 3Fi***; p=0.3479, unpaired t-test), and voltage sensitivity of $Ca^{2+}$ influx (***Figure 3—figure supplement 1B***; p=0.3470, unpaired t-test) did not differ significantly between IHCs contacted by low or high *SR* synapses. This seems to rule out a potential scenario in which the diverse SGN firing properties would be caused by varying average properties of $Ca^{2+}$ channels among different presynaptic IHCs. $Q_{50\ EPSC}$ (***Figure 3Fii***) of high *SR* synapses (–47.76±1.4 mV; n=10) was 6.76±2.0 mV more negative compared to low *SR* synapses (–41.00±1.2 mV; n=19; p=0.0021, unpaired t-test). Accordingly, high *SR* synapses had lower voltage thresholds of release than low *SR* synapses (***Figure 3Eii***, $Q_{10\ EPSC}$ of

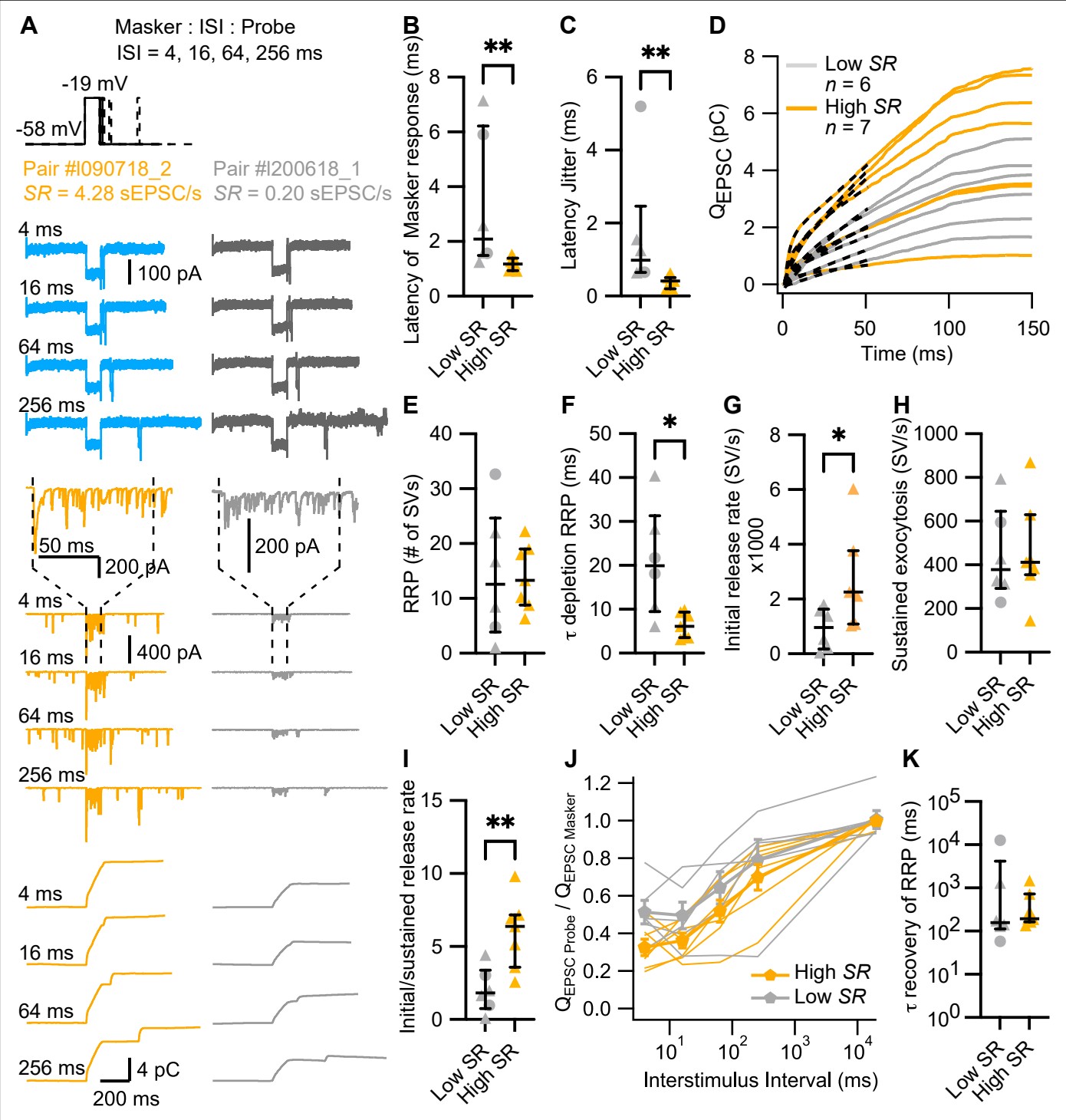

**Figure 4.** High spontaneous release synapses have shorter synaptic delay and higher initial release rates. (**A**) 'Forward masking' voltage protocol to study depletion and recovery of RRP and data of an exemplary high *SR* synapse (left panels, blue and orange) and low *SR* synapse (right panels, gray): Ca²⁺ currents (second from top, $I_{Ca}$), evoked EPSCs (eEPSCs) (second from bottom), and $Q_{EPSC}$ (bottom). The stimulus (top panel) consists of two sequential voltage steps ('masker' and 'probe') separated by different interstimulus intervals (ISI). Dashed vertical lines on top of eEPSC data indicate the onset and offset of the masker stimulus. (**B**) Latencies of the eEPSCs (eEPSC$_{onset}$ – Masker$_{onset}$) were significantly shorter in high *SR* than low *SR* synapses. (**C**) High *SR* synapses also had less latency jitter. (**D**) Pool depletion dynamics were studied by fitting the sum of a single exponential and a line function (black discontinuous line) to the first 50 ms of the average $Q_{EPSC}$ trace in response to the masker stimulus. (**E–I**) RRP, time constant ($\tau$) of depletion, initial release rate, and sustained release were calculated from the fits and the mean $Q_{sEPSC}$ for each pair. High *SR* synapses depleted the RRP with faster time constants (**F**) and reached higher initial release rates (**G**) followed by a stronger adaptation (**I**) (**J**) Recovery from RRP depletion shown

*Figure 4 continued on next page*

*Figure 4 continued*

as ratio of $Q_{EPSC\ probe}$ and $Q_{EPSC\ masker}$ (mean ± sem) during the first 10 ms of the stimulus. (**K**) Time constant of recovery from RRP depletion obtained from single exponential fits to the traces shown in J (see *Figure 4—figure supplement 1L*). Panels **B**, **C**, **E–I**, and **K** show individual data points with the median and interquartile range overlaid (line). Synapses were classified as △ pillar or ○ modiolar, and as Low *SR*<1 sEPSC/s≤High *SR*.

The online version of this article includes the following figure supplement(s) for figure 4:

**Figure supplement 1.** Parameters of synaptic vesicle pool dynamics in forward masking protocols.

–56.91±1.5 mV [median –56.64 mV] vs –47.39±1.4 mV [median –48.89 mV]; p=0.0001, Mann-Whitney U test). The hyperpolarized shift was not significant for $Q_{90\ EPSC}$ (*Figure 3—figure supplement 1D*; p=0.1706, unpaired t-test). The voltage sensitivity of release, determined by a slope factor, tended to be lower in high *SR* (4.16±0.75 mV) than in low *SR* (2.9±0.35 mV) synapses without reaching statistical significance (*Figure 3—figure supplement 1E*; p=0.0940, unpaired t-test;). The dynamic range, defined as the voltage range for which the exocytosis changes from 10% to 90% ($Q_{90\ EPSC} - Q_{10\ EPSC}$), tended to be larger for high *SR* synapses without reaching significance (*Figure 3G*; 18.30±3.3 mV for high *SR* synapses vs 12.78±1.5 mV for low *SR* synapses; p=0.0940, unpaired t-test). The voltage dependence of activation of $Ca^{2+}$ influx and of SV release did not differ significantly when the synapses were grouped based on their topographical location at the IHC (n=12 modiolar synapses vs n=17 pillar synapses; *Figure 3—figure supplement 1F–K*). However, pillar *SR* synapses had a tendency to show lower voltage thresholds of release than modiolar synapses (*Figure 3—figure supplement 1Gii*; $Q_{10\ EPSC}$ of –52.78±1.8 mV [median –53.73 mV] vs –47.68±1.8 mV [median –49.30 mV]; p=0.0725, Mann-Whitney U test). Altogether, these results demonstrate that high *SR* synapses release at more hyperpolarized voltages than low *SR* synapses.

Finally, we studied the apparent $Ca^{2+}$ dependence of SV release during the aforementioned IV protocol, i.e., in the range of IHC receptor potentials. This protocol varies $Ca^{2+}$ influx mainly via changing the channel open probability and to a lesser extent by changing the single channel current. We note that a supralinear intrinsic $Ca^{2+}$ dependence of exocytosis in IHCs (i.e. $Ca^{2+}$ cooperativity, $m$~3–4 when changing the single channel current) has been observed for IHCs of the cochlear apex in mice after hearing onset (*Brandt et al., 2005*; *Jaime Tobón and Moser, 2023*; *Özçete and Moser, 2021*; *Wong et al., 2014*). This is thought to reflect the cooperative binding of ~4 $Ca^{2+}$ ions required to trigger IHC exocytosis (*Beutner et al., 2001*). In contrast, a lower $Ca^{2+}$ cooperativity was observed in these studies when primarily changing the number of open $Ca^{2+}$ channels ($m$<2). This difference in $m$ observed for the apparent $Ca^{2+}$ dependence of exocytosis has been taken to suggest a tight, $Ca^{2+}$ nanodomain-like control of release sites by one or few $Ca^{2+}$ channel(s) in line with classical studies of the $Ca^{2+}$ dependence of transmitter release (*Augustine et al., 1991*). Here, we related changes of release at individual synapses ($\Delta Q_{EPSC}$) to the change of the integrated IHC $Ca^{2+}$ influx ($\Delta Q_{Ca}$). We fitted power functions ($Q_{EPSC} = a(Q_{Ca})^m$) to the relationships for individual synapses (*Figure 3—figure supplement 2*) and found $Ca^{2+}$ cooperativities of $m$<2 for all but 2 synapses. This result suggests a tight, $Ca^{2+}$ nanodomain-like control of release sites by one or few $Ca^{2+}$ channel(s). Interestingly, however, high *SR* synapses, on average, had significantly lower $Ca^{2+}$ cooperativities than low *SR* synapses (*Figure 3H*; $m_{highSR}$ of 0.8±0.1 [median 0.75; n=10] vs $m_{lowSR}$ of 1.4±0.1 [median 1.37; n=21]; p=0.0016, Mann-Whitney U test). The fit to pooled normalized data of high and low *SR* synapses yielded the same $Ca^{2+}$ cooperativities of $m_{highSR}$ of 0.8 and $m_{lowSR}$ of 1.4 (*Figure 3I*). When grouped based on their modiolar or pillar location, pillar synapses showed significantly lower $Ca^{2+}$ cooperativities than modiolar synapses (*Figure 3—figure supplement 1L*; $m_{pillar}$ of 1.0±0.08 [median 0.88; n=19] vs $m_{modiolar}$ of 1.6±0.2 [median 1.3; n=12]; p=0.0202, Mann-Whitney U test). Our findings indicate that most afferent IHC synapses of hearing mice employ a tight, $Ca^{2+}$ nanodomain-like control of release sites by one or few $Ca^{2+}$ channel(s) for physiological sound encoding. Yet, quantitative differences in coupling seem to exist between high *SR*/pillar synapses and low *SR*/modiolar synapses, whereby a control of SV release by ~1 $Ca^{2+}$ channel prevails at high *SR*/pillar synapses.

## SV pool dynamics at individual IHC AZs

In 13 of the 31 aforementioned paired recordings (6 classified as low *SR* and 7 as high *SR*; 2 belonging to modiolar and 11 to pillar synapses), we employed a forward masking paradigm to study SV pool dynamics of single afferent synapses. The forward masking paradigm (*Harris and Dallos, 1979*) is

commonly used for in vivo analysis of SGN spike rate adaptation and recovery from adaptation, which has been attributed to the depletion of readily releasable pool of SVs (RRP) and the recovery from depletion (*Avissar et al., 2013*; *Frank et al., 2010*; *Furukawa and Matsuura, 1978*; *Goutman, 2017*; *Goutman and Glowatzki, 2007*; *Li et al., 2009*; *Moser and Beutner, 2000*; *Schroeder and Hall, 1974*). Typically, the in vivo protocol is applied at saturating sound pressure levels, which we aimed to mimic using strong step IHC depolarizations (to –19 mV from –58 mV) separated by different inter-stimulus intervals (ISI: 4, 16, 64, and 256 ms) (*Figure 4A*). In analogy to the in vivo forward masking paradigm, the first stimulus - called *masker*, as it depresses the response to a subsequent stimulus when applied in rapid succession - had a duration of 100 ms. The second stimulus - denominated *probe* - lasted for 15 ms. The recordings included a time frame of 400 ms preceding the 'masker' and 400 ms following the 'probe', and the interval between masker and masker was 20 s. Applied to recordings of eEPSCs, the forward masking protocol provides experimental access to the initial RRP release rates, kinetics, and extent of RRP depletion, sustained exocytosis, as well as recovery from RRP depletion. To accommodate the stochasticity of SV release from the RRP of IHC AZs, we run each protocol several times (≥3 to ≤20), which is routinely done for in vivo SGN physiology, but challenging ex vivo given the fragile and typically short-lived paired pre- and postsynaptic recordings (e.g. *Goutman, 2017*; *Goutman and Glowatzki, 2007*). Note that we did not employ cyclothiazide to inhibit AMPA receptor desensitization and reduce its contribution to postsynaptic eEPSC depression (*Goutman, 2017*), given the potential presynaptic effects of cyclothiazide in synaptic release (*Diamond and Jahr, 1995*; *Dittman and Regehr, 1998*).

For the analysis of evoked release dynamics, we focused on the response evoked by the masker. At the presynaptic level, there was no difference in the peak, initial and final IHC $Ca^{2+}$ currents ($I_{Ca}$) and $Ca^{2+}$ current charge ($Q_{Ca}$) between the recordings from high and low *SR* synapses (*Figure 4—figure supplement 1A–D*). We calculated the synaptic delay from the onset of the masker to the onset of the eEPSC. High *SR* synapses had significantly shorter synaptic delays (*Figure 4B*), with mean latencies of the first eEPSC after stimulus onset of 1.17±0.09 ms (median 1.18) compared to 3.34±1.04 ms (median 2.09) in low *SR* pairs (p=0.0082, Mann-Whitney U test). This result was corroborated in a bigger sample size, when we compared the synaptic latencies of the 31 pairs to 10 ms pulses to –19 mV (*Figure 4—figure supplement 1E*; latency of 1.19±0.14 ms [median 1.14] in high *SR* compared to 2.57±0.48 ms [median 1.79] in low *SR* synapses; p=0.0101, Mann-Whitney U test). The latency jitter (measured as the standard deviation of the masker-evoked EPSC latencies; *Figure 4C*) was also significantly smaller in high *SR* synapses compared to low *SR* synapses (0.38±0.07 ms [median 0.41] vs 1.67±0.72 ms [median 0.98], respectively; p=0.0012, Mann-Whitney U test). Additionally, high *SR* synapses had significantly larger peak amplitudes of the masker-evoked EPSCs which reports the initial release from the RRP (*Figure 4—figure supplement 1F*; –420.9±54.22 pA for high *SR* synapses vs –240.1±23.19 pA for low *SR* synapses; p=0.0149, unpaired t-test). In contrast, the charge of masker-evoked EPSCs ($Q_{EPSC\ MASKER}$) was more comparable (*Figure 4—figure supplement 1G*; 4.512±0.82 pC for high SR vs 3.020±0.43 pC for low *SR* pairs; p=0.1535, unpaired t-test). We fitted the first 50 ms of the $Q_{EPSC\ MASKER}$ with the sum of a single exponential and a line function in order to analyze RRP depletion and sustained release (*Figure 4—figure supplement 1H*; dashed lines in *Figure 4D*). The amplitude of the exponential component (A1), thought to reflect RRP exocytosis, was not different between high and low *SR* synapses (*Figure 4—figure supplement 1I*; p=0.4092, unpaired t-test). Likewise, the slope of the linear component, reflecting sustained exocytosis, did not differ significantly between the two groups (*Figure 4—figure supplement 1J*; p=0.1807, Mann-Whitney U test).

To quantify synaptic release in terms of SVs, we divided $Q_{EPSC\ MASKER}$ by the mean $Q_{sEPSC}$ recorded for each pair (see *Figure 2K*). This builds on our assumption that each sEPSC corresponds to a unitary release event ('univesicular mode of release'; *Chapochnikov et al., 2014*; *Grabner and Moser, 2018*; *Huang and Moser, 2018*). The quantal content (RRP size in #SV) was comparable between high and low *SR* synapses (*Figure 4E*, 13.99±2.239 SVs vs 14.28±4.839 SVs, respectively; p=0.9553, unpaired t-test). However, the RRP depleted significantly faster in high *SR* synapses (*Figure 4F*): $\tau_{RRP\ depletion}$ was 6.347±1.096 ms (median 6.13) for high *SR* vs 20.88±5.063 ms (median 19.93) for low *SR* synapses (p=0.0140, Mann-Whitney U test). Accordingly, high *SR* synapses showed significantly higher initial release rates compared to low *SR* synapses (*Figure 4G*; 2651±660.0 SV/s vs 927.4±307.4 SV/s; p=0.0472, unpaired t-test), which given the comparable RRP size, indicates a higher release probability of the high *SR* synapses. Moreover, high *SR* synapses showed stronger depression of the release

rate (*Figure 4I*; the ratio of initial/sustained release rate was 5.941±0.916 for high *SR* pairs and 2.023±0.6251 for low *SR* pairs; p=0.006, unpaired t-test), despite comparable sustained release rates (*Figure 4H*; p=0.9258, unpaired t-test).

Finally, we determined RRP recovery from depletion using the ratio $Q_{EPSC\ Probe}/Q_{EPSC\ Masker}$, whereby $Q_{EPSC}$ for masker and probe was estimated for the first 10 ms of stimulation. *Figure 4J* plots the ratio for each ISI, including the masker-to-masker interval. All the synapses exhibited a smaller response (*Figure 4J*) and a longer response latency (*Figure 4—figure supplement 1K*) to the probe compared to the response to the masker stimulus. This short-term synaptic depression probably reflects RRP depletion by the masker stimulus (*Avissar et al., 2013*; *Cho et al., 2011*; *Frank et al., 2010*; *Moser and Beutner, 2000*), in contrast to previous reports of synaptic facilitation when employing short paired stimuli from more hyperpolarized resting potential that do not trigger the full RRP release (*Chen and von Gersdorff, 2019*; *Goutman and Glowatzki, 2011*). Surprisingly, and contrary to a previous report in auditory bullfrog synapses (*Cho et al., 2011*), approximately half of the synapses showed a lower ratio at 16 ms than at 4 ms regardless of their *SR*. Therefore, to determine the kinetics of recovery from RRP depletion, we fitted a single exponential to the recovery data over the ISI range from 16 ms to 20 s (*Figure 4—figure supplement 1L*). We did not find a significant difference between high *SR* and low *SR* synapses (*Figure 4K*; $\tau_{recovery\ RRP}$: 451.5±187.0 ms [median 192.7] vs 2416±2064 ms [median 157.9], respectively; p=0.5338, Mann-Whitney U test). In 4 high *SR* and 4 low *SR* synapses, spontaneous activity was resumed shortly after the offset of the probe. The time to this first sEPSC took longer in high *SR* synapses compared to low ones (*Figure 4—figure supplement 1M*; 185.2±25.25 ms [median 161.6] vs 104.5±24.10 ms [median 108.4], respectively; p=0.0286, Mann-Whitney U test), again indicating stronger synaptic depression on high *SR* synapses.

## Discussion

Much of the information on synaptic sound encoding at afferent IHC-SGN synapses has been obtained from either juxtacellular recordings of SGN firing in vivo or from ex vivo patch-clamp recordings. Yet, it has remained difficult to reconcile those in vivo and ex vivo results and to establish a unified account of sound intensity coding in the auditory nerve given differences in experimental conditions, animal models, and protocols employed. Here, we biophysically characterized the heterogeneous function of afferent SGN synapses in hearing mice with reference to their rate of spontaneous transmission (*SR*) as a surrogate of SGN SR that informs their functional properties. We performed paired pre- and post-synaptic patch-clamp recordings from single IHC synapses of hearing mice under near-physiological conditions using protocols adapted from in vivo characterization of SGN's response properties. Using this approach, we were able to distinguish synapses with low and high *SR*, which we propose to provide the input into low and high SR SGNs. We found that about 90% of high *SR* synapses were located at the pillar side of the IHC. High *SR* synapses had larger sEPSCs with a monophasic (or more compact) waveform, lower voltage thresholds of release, shorter synaptic delays, tighter coupling of release sites to $Ca^{2+}$ channels, as well as higher initial release rates and shorter RRP depletion time constants. RRP size, rate of sustained exocytosis, and kinetics of RRP recovery from depletion were comparable between high and low *SR* synapses. We conclude that high *SR* synapses exhibit higher release probability which likely reflects the tighter coupling of $Ca^{2+}$ channels and release sites. This diversity in the response properties of individual synapses most likely expands the capacity of a single IHC to encode sound intensity over the wide range of audible sound pressures.

### Diversity of spontaneous release and their topographical segregation

The *SR* range observed in our paired recordings from mouse afferent synapses (0–18 sEPSC/s) agrees with results obtained without patch-clamping the IHC (0–16 sEPSC/s) and with previous ex vivo reports using loose patch recordings from SGNs of p15–17 rats (0.1–16.42 spikes/s; *Wu et al., 2016*). However, the maximum rate is considerably smaller than those recorded in vivo from single SGNs of p14–15 mice (up to 60 spikes/s; *Wong et al., 2013*). This threefold difference between ex vivo and in vivo recordings of the same age group could indicate that, in vivo, the IHC resting potential might be more depolarized and/or subject to spontaneous fluctuations that can trigger $Ca^{2+}$ channel openings and release. Additionally, the presence of $K^{+}$ channel blockers (tetraethylammonium [TEA] and $Cs^{+}$)

and differences in pH could all have an impact on the excitability of the cell and the kinetics of the cellular processes.

Other important factors might explain the higher SR reported in vivo: (i) The intrinsic biophysical properties of SGNs which could further expand the firing rate distribution (*Markowitz and Kalluri, 2020*), for instance, due to spikes initiated intrinsically and not activated by an EPSP (*Wu et al., 2016*); (ii) a sampling bias of synapses with *SR* above 20 sEPSC/s given that they constitute less than 35% of SGNs from 8- to 17-week-old mice (*Taberner and Liberman, 2005*); and (iii) the developmental recruitment of high SR SGNs with age (*Niwa et al., 2021*; *Romand, 1984*; *Walsh and McGee, 1987*; *Wong et al., 2013*; *Wu et al., 2016*). 90% of the paired recordings (and 60% for the bouton recordings) of our dataset were obtained from mice between p14 and p17, in which spontaneous activity is still low compared to older age groups (p19–21: 0–44.22 spikes/s; p29–32: 0.11–54.9 spikes/s [*Wu et al., 2016*]; p28: 0–47.94 spikes/s [*Siebald et al., 2023*]).

In analogy to the pioneering finding of a synaptic segregation of cat SGNs according to SR along the pillar-modiolar axis of IHCs (*Liberman, 1982*; *Merchan-Perez and Liberman, 1996*), we found that about 90% of high *SR* were located on the pillar IHC side. Yet, not all the synapses of the pillar IHC side had high *SR*, which agrees with a recent study of molecularly tagged SGNs (*Siebald et al., 2023*). These findings suggest that high-frequency and large-amplitude sEPSCs occur predominantly in synapses with smaller ribbons and AZs, opposing results from retinal cells in which smaller ribbons resulted in reduced frequency and amplitude of EPSCs (*Mehta et al., 2013*). It is important to point out that our modiolar/pillar classification is less precise than that of other studies in which the synapse position was quantitatively assigned (*Frank et al., 2009*; *Liberman et al., 2011*; *Ohn et al., 2016*; *Özçete and Moser, 2021*). Moreover, other studies also support an overall pillar-modiolar gradient with 'salt and pepper' intermingling of synaptic properties rather than their strict segregation (*Ohn et al., 2016*; *Özçete and Moser, 2021*).

Similar to previous reports (e.g. *Chapochnikov et al., 2014*; *Glowatzki and Fuchs, 2002*; *Grant et al., 2010*; *Huang and Moser, 2018*; *Rutherford et al., 2012*), we found a high variability in the waveform and amplitude of sEPSCs between synapses, which apparently do not strictly depend on the topographical location of the synapse (the present work and *Niwa et al., 2021*). Yet, high *SR* synapses had larger and monophasic (or temporally more compact) sEPSCs. This likely also explains the shorter 10–90% rise time of the sEPSC in high *SR* synapses, as monophasic sEPSC have shorter rise times than the multiphasic ones (*Chapochnikov et al., 2014*; *Glowatzki and Fuchs, 2002*; *Huang and Moser, 2018*). The difference in the percentage of monophasic sEPSCs and rise times in low and high *SR* synapses could arise from variability in the fusion pore dynamics on the way to SV fusion and/or on the number of SVs released in timely manner. In the view of the multivesicular hypothesis of spontaneous release (*Glowatzki and Fuchs, 2002*; *Li et al., 2009*; *Niwa et al., 2021*; *Schnee et al., 2013*), SVs of a modiolar AZ might fuse in an uncoordinated manner, creating EPSCs with a less compact waveform and slower rise times.

Alternatively and our favorite hypothesis, each sEPSC corresponds to a unitary release event ('univesicular mode of release'; *Chapochnikov et al., 2014*; *Grabner and Moser, 2018*; *Huang and Moser, 2018*; *Young et al., 2021*) that has the capacity to drive action potential firing (*Rutherford et al., 2012*). In the framework of the univesicular hypothesis of spontaneous release, the flickering of the fusion pore, prior to or instead of full collapse fusion of the SV, might be favored in low *SR* synapses, leading to slower sEPSC rise times and a lower percentage of monophasic sEPSCs. Such heterogeneity of fusion pore dynamics has been reported in chromaffin cells, calyx of Held, and hippocampal neurons (*Chang et al., 2021*; *Henkel et al., 2019*; *Shin et al., 2020*; *Shin et al., 2018*).

Finally, we note that sEPSC amplitudes of IHC synapses in hearing mice (present study and *Niwa et al., 2021*) seem lower than in previous ex vivo studies on IHC synapses of hearing rats (*Chapochnikov et al., 2014*; *Grant et al., 2010*; *Huang and Moser, 2018*; *Young et al., 2021*). In rats, the EPSC amplitude distribution changes with maturation, from highly skewed to the left with a peak around –30 pA to a Gaussian-like distribution with a peak at –375 pA (*Grant et al., 2010*). This does not seem to be the case in mouse IHC synapses. Average EPSC amplitudes in pre-hearing mice are around –100 to –150 pA (*Chapochnikov et al., 2014*), even with 40 mM K⁺ stimulation (*Jing et al., 2013*; *Takago et al., 2018*). On the contrary, mean EPSC amplitudes in hearing mice remained small (around –100 pA) in resting conditions (*Niwa et al., 2021* and the present study), but became

significantly larger upon stimulation with 40 mM K$^+$ (*Niwa et al., 2021*) or voltage depolarizations (the present study, *Figure 1—figure supplement 1G*).

## Candidate mechanisms distinguishing evoked release at low and high *SR* synapses

The temporal and quantal resolution offered by paired recordings allowed us to analyze the biophysical properties of evoked synaptic transmission in relation to the *SR* of the given synapse. In an intriguing resemblance with in vivo evoked firing properties of high SR SGNs (*Bourien et al., 2014*; *Buran et al., 2010*; *Relkin and Doucet, 1991*; *Rhode and Smith, 1985*; *Taberner and Liberman, 2005*), high *SR* synapses showed lower voltage (~sound pressure in vivo) thresholds of synaptic transmission (~firing in vivo), shorter and less variable synaptic latencies (~first spike latencies in vivo), and higher initial release rates (~onset firing rate in vivo). In addition, we found stronger synaptic depression at high *SR* synapses, which agrees well with the finding of a greater ratio of peak to adapted firing rate in high SR SGNs recorded in vivo (*Taberner and Liberman, 2005*). These results support the hypothesis that IHC synaptic heterogeneity (*Frank et al., 2009*; *Hua et al., 2021*; *Ohn et al., 2016*; *Özçete and Moser, 2021*; *Reijntjes et al., 2020*) contributes to the diversity of spontaneous and sound-evoked SGN firing.

How do high *SR* synapses with likely smaller ribbons and lower maximal Ca$^{2+}$ influx achieve a shorter latency and higher initial release rate? Our hypothesis is that, in high *SR* synapses, a more hyperpolarized Ca$^{2+}$ channel activation (*Ohn et al., 2016*) in combination with tighter coupling between the Ca$^{2+}$ channels and the Ca$^{2+}$ sensor of fusion (this work and *Özçete and Moser, 2021*) would enable a faster response with a greater initial SV release probability for a given stimulus. The spatial coupling of the Ca$^{2+}$ channel to the SV release site has also been shown to greatly affect release probability in other synapses (*Eggermann et al., 2011*; *Fekete et al., 2019*; *Moser et al., 2020*; *Rebola et al., 2019*). Thus, a heterogenous Ca$^{2+}$ coupling would diversify the response properties (i.e. SV release probability) of individual synapses to the same stimulus. In IHCs, this is particularly important for sound intensity and temporal coding (reviewed in *Moser et al., 2023*). Interestingly, genetic disruptions that shifted the voltage dependence had a greater impact on the in vivo distribution of SR and onset firing rate of SGNs than mutations that changed the maximal synaptic Ca$^{2+}$ influx (*Jean et al., 2018*; *Ohn et al., 2016*).

Paired pre- and postsynaptic patch-clamp recordings (this work) and single synapse imaging of presynaptic Ca$^{2+}$ signals and glutamate release (*Özçete and Moser, 2021*) jointly found a lower apparent Ca$^{2+}$ cooperativity in pillar synapses during depolarizations within the range of receptor potentials. The sensitivity and temporal resolution of paired recordings further allowed us to classify the synapses based on their spontaneous rate and support the hypothesis that high SR SGNs receive input from AZ with tighter coupling than low SR SGNs. However, single synapse imaging (*Özçete and Moser, 2021*) found a wider range of apparent Ca$^{2+}$ cooperativities than our two non-overlapping datasets for paired patch-clamp recordings (this work and *Jaime Tobón and Moser, 2023*). This might reflect two important technical differences: (i) single synapse imaging assessed the presynaptic Ca$^{2+}$ influx of the specific synapse, while in paired recordings we related release to the whole-cell Ca$^{2+}$ influx, and (ii) the temporal resolution of paired recordings allowed to study the initial release rate using shorter stimuli than in imaging, which avoids an impact of RRP depletion and ongoing SV replenishment. Future studies, potentially combining paired patch-clamp recordings with imaging of presynaptic Ca$^{2+}$ signals, will be needed to further scrutinize the heterogeneity of Ca$^{2+}$ dependence of release in IHCs and its impact on release probability.

Other factors that affect release probability include variations in the number of open Ca$^{2+}$ channels at the AZ (*Gratz et al., 2019*; *Holderith et al., 2012*; *Scimemi and Diamond, 2012*; *Sheng et al., 2012*; *Wong et al., 2013*) and the fusion competence of the SV (*Klenchin and Martin, 2000*), including the priming and docking state (*Lin et al., 2022*; *Neher and Brose, 2018*). The ensuing Ca$^{2+}$ influx in pillar synapses might facilitate Ca$^{2+}$ channels and priming of SVs (*Cho and von Gersdorff, 2012*; *Goutman and Glowatzki, 2011*; *Goutman and Glowatzki, 2007*; *Michalski et al., 2017*; *Moser and Beutner, 2000*; *Pangršič et al., 2015*; *Schnee et al., 2011*; *Spassova et al., 2004*) and contribute to the observed results in high *SR* synapses. Regarding the fusion competence of SVs, it is unknown whether modiolar and pillar synapses exhibit different numbers of docked and primed SVs. To date, ultrastructural studies that resolve docked and tethered SVs have not addressed the

topographical location of the AZ in the murine IHC (*Chakrabarti et al., 2022*; *Chakrabarti et al., 2018*).

Besides SV release probability, RRP size co-determines neurotransmitter release. Our estimated RRP of about 14 SVs in both high and low *SR* synapses compares well to prior estimates obtained using ex vivo electrophysiology (10–40 SVs: *Goutman and Glowatzki, 2007*; *Jean et al., 2018*; *Johnson et al., 2005*; *Khimich et al., 2005*; *Moser and Beutner, 2000*; *Pangrsic et al., 2010*; *Schnee et al., 2005*), model-based analysis of SGN firing (4–40 SVs; *Frank et al., 2010*; *Jean et al., 2018*; *Peterson et al., 2014*), and electron microscopy (10–16 SVs within 50 nm of the presynaptic membrane; *Chakrabarti et al., 2018*; *Frank et al., 2010*; *Graydon et al., 2011*; *Kantardzhieva et al., 2013*; *Khimich et al., 2005*; *Pangrsic et al., 2010*). However, previous reports based on electron microscopy (*Kantardzhieva et al., 2013*; *Merchan-Perez and Liberman, 1996*; *Michanski et al., 2019*) suggested larger pools of SVs at modiolar synapses, while our electrophysiological estimate of RRP size was comparable between low and high *SR* synapses. This finding argues against a strong contribution of RRP size to the observed differences in neurotransmitter release. However, higher release probability with comparable RRP size explains higher initial release rates, which likely explain the faster and temporally more precise postsynaptic depolarization that is likely to turn into shorter first spike latencies and lower first spike latency jitter (this study and *Buran et al., 2010*).

Finally, the heterogeneity in the functional properties of IHC synapses could arise from molecular heterogeneity of the AZ. In central glutamatergic synapses, molecular heterogeneity of synaptic proteins plays a critical role in the modulation of SV release probability and priming state (*Neher and Brose, 2018*; *Wichmann and Kuner, 2022*). For instance, differential isoforms of priming factors and scaffold proteins have been suggested to tune the functional synaptic diversity of central synapses (*Fulterer et al., 2018*; *Rebola et al., 2019*; *Rosenmund et al., 2002*). Cochlear IHCs have an unconventional fusion machinery that appears to work without neuronal SNARES (*Nouvian et al., 2011*) (but see *Calvet et al., 2022*) and priming factors such as Munc13 and CAPS (*Vogl et al., 2015*). Therefore, future studies will need to determine the molecular nanoanatomy underlying the specific AZ nanophysiology and functional synaptic heterogeneity at IHCs. Promising candidates include RBPs (*Butola et al., 2021*; *Grauel et al., 2016*; *Krinner et al., 2021*; *Krinner et al., 2017*; *Petzoldt et al., 2020*), RIMs (*Jung et al., 2015*; *Picher et al., 2017b*), and Septin (*Fekete et al., 2019*; *Yang et al., 2010*).

## Challenges for relating synaptic and neural response properties

Next to providing support for the presynaptic hypothesis of functional SGN diversity, the present study also highlights some of the challenges met when aiming to bridge the gap between presynaptic hair cell function and neural sound encoding. Despite major efforts undertaken to match experimental conditions and protocols, it remains difficult to reconcile some findings of ex vivo and in vivo physiology. Parameters such as RRP size (~# spikes of the rapidly adapting component of firing), sustained exocytosis (~adapted firing rate in vivo), recovery of spontaneous and evoked release (~recovery from forward masking in vivo) did not differ among our high and low *SR* synapses, and contrasts with in vivo data (e.g. *Bourien et al., 2014*; *Buran et al., 2010*; *Relkin and Doucet, 1991*; *Rhode and Smith, 1985*; *Taberner and Liberman, 2005*).

Of particular interest is that the dynamic ranges and slope of release-intensity relationship of high and low *SR* synapses diverge from the expectations if assuming that high SR SGNs are driven by high *SR* synapses. High *SR* synapses tended to show broader dynamic ranges with shallower slopes, while, in vivo, high SR SGNs show smaller dynamic ranges and steeper slopes than the low SR ones (*Ohlemiller et al., 1991*; *Winter et al., 1990*). Could this reflect the non-linear saturating properties of the basilar membrane (*Sachs et al., 1989*; *Sachs and Abbas, 1974*; *Yates et al., 1990*) (discussed in *Ohlemiller et al., 1991*) which might widen the rate level function of low SR SGNs? Or is it due to a partial depletion of the 'standing' RRP (i.e. the occupancy of the RRP release sites with a fusion-competent SV; *Moser, 2020*; *Pangrsic et al., 2010*) at high *SR* synapses in vivo? It remains to be determined whether this and the other aforementioned differences between our data and in vivo reports could be attributed to mechanisms downstream of glutamate release and AMPA receptor activation. The possible mechanisms include but are not limited to: (i) different spike rates due to diverse EPSC waveforms (*Rutherford et al., 2012*); (ii) differences in SGN excitability (*Crozier and Davis, 2014*; *Markowitz and Kalluri, 2020*; *Smith et al., 2015*) due to heterogenous molecular (*Petitpré et al., 2020*; *Petitpré et al., 2018*; *Shrestha et al., 2018*; *Sun et al., 2018*) and morphological

profiles (*Liberman, 1980*; *Merchan-Perez and Liberman, 1996*; *Tsuji and Liberman, 1997*); and (iii) differences in efferent innervation of SGNs (*Hua et al., 2021*; *Liberman, 1990*; *Ruel et al., 2001*; *Wu et al., 2020*; *Yin et al., 2014*). Certainly, caution is to be applied for the comparison of ex vivo and in vivo data due to the partial disruption of the physiological milieu despite our efforts to maintain near-physiological conditions, and the incomplete synaptic maturation when focusing ex vivo experiments on the third postnatal week soon after hearing onset.

Clearly more work is needed to elucidate the mechanisms of SGN firing diversity in the cochlea. Ideally, future studies will combine in vivo and ex vivo experiments, such as combining physiological SGN characterization with neural backtracing and synaptic morphology of labeled SGNs using volume imaging of afferent and efferent connectivity (*Hua et al., 2021*). Moreover, combining optogenetic IHC stimulation with imaging of SGN activity could provide higher throughput and serve post hoc morphology. Finally, paired patch-clamp recordings, as done in the present study, could be combined with SGN subtype-specific molecular labeling, fiber tracing, and immunolabeling to further relate synaptic transmission and SGN neurophysiology.

# Materials and methods

**Key resources table**

| Reagent type (species) or resource | Designation | Source or reference | Identifiers | Additional information |
|---|---|---|---|---|
| Strain, strain background (*Mus musculus*) | C57BL/6N | Jackson Laboratory (https://www.jax.org) | RRID:IMSR_JAX:005304 | |
| Software, algorithm | Patchmaster | HEKA Elektronik, (http://www.heka.com/products/products_main.html#soft_pm) | RRID:SCR_000034: | |
| Software, algorithm | Igor Pro software package | WaveMetrics (http://www.wavemetrics.com/products/igorpro/igorpro.htm) | RRID:SCR_000325 | |
| Software, algorithm | Patchers Power Tools | Igor Pro XOP (http://www3.mpibpc.mpg.de/groups/neher/index.php?page=software) | RRID:SCR_001950 | |
| Software, algorithm | NeuroMatic | ThinkRandom (http://www.neuromatic.thinkrandom.com/) | RRID:SCR_004186 | |
| Software, algorithm | Excel | Microsoft (https://www.microsoft.com/en-gb/) | RRID:SCR_016137 | |
| Software, algorithm | GraphPad Prism software | GraphPad Prism (https://graphpad.com) | RRID:SCR_002798 | |
| Software, algorithm | Adobe Illustrator | Adobe (http://www.adobe.com/products/illustrator.html) | RRID:SCR_010279 | |

## Animals and tissue preparation

c57BL/6N mice of either sex between p14 and p27 were used. For paired recordings, the number of animals per age was: p14 (9), p15 (7), p16 (9), p17 (5), p18 (2), p20 (1). For the bouton recordings of *Figure 2—figure supplement 1A*, the number of animals per age was: p14 (2), p15 (3), p16 (3), p21 (1), p24 (1), p25 (1), p27 (1). The animal handling and euthanizing complied with national animal care guidelines and were announced to the local animal welfare committee of the University of Göttingen and the Max Planck Institute for Multidisciplinary Sciences, as well as to the Animal Welfare Office of the State of Lower Saxony, Germany (announcement T 37.03). Animals were sacrificed by decapitation and the cochleae were extracted in modified Hepes Hank's solution containing: 5.36 mM KCl, 141.7 mM NaCl, 1 mM MgCl$_2$-6H$_2$O, 0.5 mM MgSO$_4$-7H$_2$O, 10 mM HEPES, 0.5 mg/ml L-glutamine, and 1 mg/ml D-glucose (pH 7.2, osmolarity of ~300 mOsm). The apical coil of the organ of Corti was dissected and placed under a grid in the recording chamber. Pillar or modiolar supporting cells were removed using soda glass pipettes in order to gain access to the basolateral face of the IHCs and to the postsynaptic boutons of type I SGNs. Dissection of the organ of Corti and cleaning of the supporting cells were performed at room temperature (20–25°C).

## Electrophysiological recordings

Pre- and postsynaptic paired patch-clamp recordings were performed at near physiological temperature (32–37°C) using an EPC-9 amplifier (HEKA electronics) (*Figure 1*). Patch electrodes were positioned

using a PatchStar micromanipulator (Scientifica, UK). Whole-cell recordings from IHCs were achieved using the perforated-patch-clamp technique (*Moser and Beutner, 2000*) using Sylgard-coated 1.5 mm borosilicate pipettes with typical resistances between 3.5 and 6 MΩ. The IHC pipette solution contained: 129 mM Cs-gluconate, 10 mM TEA-Cl, 10 mM 4-AP, 10 mM HEPES, 1 mM $MgCl_2$ (pH 7.2, osmolarity of ~290 mOsm), as well as 300 µg/ml amphotericin B added prior to the experiment. Once the series resistance of the IHC reached below 30 MΩ, whole-cell voltage-clamp recordings from a contacting bouton was established as described in previous studies (*Glowatzki and Fuchs, 2002*; *Grant et al., 2011*; *Huang and Moser, 2018*). For two pairs, the bouton recording was established first and then the IHC. Sylgard-coated 1.0 mm borosilicate pipettes with typical resistances between 7 and 12 MΩ were used for the postsynaptic recordings. The bouton pipette solution contained: 137 mM KCl, 5 mM EGTA, 5 mM HEPES, 1 mM $Na_2$-GTP, 2.5 mM $Na_2$-ATP, 3.5 mM $MgCl_2 \cdot 6H_2O$, and 0.1 mM $CaCl_2$ (pH 7.2 and osmolarity of ~290 mOsm). The organ of Corti was continuously perfused with an extracellular solution containing 4.2 mM KCl, 95–100 mM NaCl, 25 mM $NaHCO_3$, 30 mM TEA-Cl, 1 mM Na-pyruvate, 0.7 mM $NH_2PO_4 \cdot H_2O$, 1 mM CsCl, 1 mM $MgCl_2 \cdot H_2O$, 1.3 mM $CaCl_2$, and 11.1 mM D-glucose (pH 7.3, osmolarity of ~310 mOsm). 2.5 µM tetrodotoxin (Tocris or Santa Cruz) was added to block voltage-gated $Na^+$ channels in the postsynaptic bouton.

Data were acquired using the Patchmaster software (HEKA electronics). The current signal was filtered at 5–10 kHz and sampled at 20–50 kHz. IHCs were voltage-clamped at a holding potential of –58 mV, around the presumed in vivo resting potential (*Johnson, 2015*). The bouton was held at a potential of –94 mV. All reported potentials were corrected for the liquid junction potential (19 mV for the IHC and 4 mV for the bouton), measured experimentally. $Ca^{2+}$ current recordings were corrected for the linear leak current using a P/n protocol. We excluded IHCs and boutons with leak currents exceeding –60 pA and –100 pA at holding potential, respectively. Average IHC series resistance ($R_s$) was 14.7±0.8 MΩ (14.63±1.04 MΩ for high *SR* synapses vs 14.83±1.07 MΩ for low *SR*; p=0.7433, Mann-Whitney U test). Average IHC membrane capacitance ($C_m$) was 8.8±0.19 pF (8.8±0.25 pF for recordings of high *SR* synapses vs 8.8±0.25 pF for low *SR*; p=0.6237, Mann-Whitney U test). The apparent series resistance of the bouton was calculated from the capacitive transient in response to a 10 mV test pulse. The actual $R_s$ was offline calculated as reported in *Huang and Moser, 2018*. Briefly, we fitted the decay phase of the capacitive transient with a double exponential. Average bouton $R_s$ from paired recordings was 64.6±3.3 MΩ (60.2±5.3 MΩ for high *SR* synapses vs 57.8±3.6 MΩ for low *SR*; p=0.7115, unpaired t-test). (Average $R_s$ from the bouton recordings from *Figure 2—figure supplement 1* was 57±4.7 MΩ). Bouton capacitance was estimated from the area under the fast component of the double exponential fit. Average bouton $C_m$ was 1.7±0.09 pF (1.8±0.19 pF for high *SR* synapses vs 1.7±0.11 pF for low *SR*; p=0.6575, unpaired t-test). Average bouton membrane resistance ($R_m$) was 1491±133.2 MΩ (1499±193.3 MΩ for high *SR* synapses vs 1487±174.3 MΩ for low *SR*; p=0.7143, Mann-Whitney U test). The proper-ties of the recordings (i.e. amplitude of EPSC) were not correlated with the bouton $R_s$ (*Figure 1—figure supplement 1*).

The threshold for sEPSC detection was four times SD of the baseline. Spontaneous activity was calculated from time windows without stimulation with the IHC held at –58 mV; either from a 5–10 s recording or by averaging the number of events from the segments before and after a depolarizing pulse (*Figure 1B*, *Supplementary file 1*). To study the depletion and recovery of the pool of vesicles, we used a protocol adapted from the forward masking protocol performed during in vivo extracel-lular recordings of SGNs (*Harris and Dallos, 1979*; *Jean et al., 2018*). It consisted of two consecutive depolarizing pulses to the voltage that elicited the highest peak of $Ca^{2+}$ current (–19 mV; *Figure 1C*). The first pulse, called masker, lasted 100 ms and it was followed by a second pulse, called probe, which lasted 15 ms. The two pulses were separated by intervals without depolarization (ISI) that lasted 4, 16, 64, and 256 ms. The waiting time between masker and masker was 20 s and each protocol was repeated between 3 and 20 times. To study the dynamic voltage range of synaptic transmission, we used a current-voltage (IV) protocol with 10 ms pulses of increasing voltage (from –70 mV/–60 mV to 70 mV in 5 mV steps). The interval between two stimuli was 1.5 s.

## Data analysis

Electrophysiological data was analyzed using the IgorPro 6 Software Package (WaveMetrics), GraphPad Prism 9 and Excel. $Ca^{2+}$ charge ($Q_{Ca}$) and EPSC charge ($Q_{EPSC}$) were estimated by taking the integral of

the current. Kinetics of sEPSCs, such as amplitude, 10–90% rise time, time constant of decay ($\tau_{decay}$), and FWHM, were calculated with Neuromatic (**Rothman and Silver, 2018**).

To obtain IV curves, we averaged the evoked $Ca^{2+}$ currents ($I_{Ca}$) during 10 ms after the start of each depolarization. Fractional activation of the $Ca^{2+}$ channels was obtained from the normalized chord conductance, $g$,

$$g = \frac{I}{(V - V_{rev})}$$

where $V$ is the membrane potential and $V_{rev}$ is the reversal potential determined by fitting a line function between the voltage of $I_{Ca\ peak}$+10 mV and the maximal depolarization. The activation curve was approximated by a first-order Boltzmann equation:

$$g = \frac{g_{max}}{1 + \exp\left(\frac{V_{half}\ I_{Ca} - V}{S}\right)}$$

where $g_{max}$ is the maximum chord conductance, $V_{half}\ I_{Ca}$ is the membrane potential at which the conductance is half activated, and $S$ is the slope factor describing the voltage sensitivity of activation.

Release intensity curves were obtained by calculating $Q_{EPSC}$ by the end of each depolarization step and fitted using a sigmoidal function:

$$Q = \frac{Q_{max}}{1 + \exp\left(\frac{Q_{50\ EPSC} - V}{rate}\right)}$$

where $Q_{max}$ is the maximal $Q_{EPSC}$ (normalized to 1), $Q_{50\ EPSC}$ corresponds to the voltage of half-maximal release (or $V_{h\ EPSC}$), and $Q$ is the EPSC charge. The dynamic range was determined as the voltage range between 10% and 90% of the maximal vesicle release. For statistical analysis of dynamic range, we included only pairs for which both the $Ca^{2+}$ fractional activation and the rate level curves were possible to fit.

The apparent $Ca^{2+}$ dependence of neurotransmitter release was studied from the 10 ms step depolarizations of the IV curves. The resulting $Q_{EPSC}$ vs IHC $Q_{Ca}$ plots from each individual pair were fitted with a power function:

$$Q_{EPSC} = a\left(Q_{Ca}\right)^{m}$$

where $m$ corresponds to the $Ca^{2+}$ cooperativity. Some pairs showed a clear saturation of release at high IHC $Q_{Ca}$. In these cases, the fit was restricted to the data points before the plateau, which was determined by visual inspection. For the pooled data, the power function was fitted to the normalized $Q_{EPSC}$ vs normalized $Q_{Ca}$. For the pairs with saturation of release, $Q_{Ca}$ was normalized to a point before the plateau.

For forward masking experiments, the postsynaptic response was averaged for all the repetitions for each paired recording (between 3 and 20, depending on the stability of the pair). Single AZ pool dynamics were determined by fitting an exponential plus line function to the first 50 ms of the average $Q_{EPSC}$ trace in response to the masker stimulus for each ISI,

$$y_0 + A1\left(1 - exp\left\{\frac{-(x - x_0)}{\tau}\right\}\right) + (x - x_0)\ slope$$

where $A1$ is the amplitude of the exponential component, $\tau$ is the time constant of the exponential component. RRP size (in SVs) was estimated from dividing $A1$ by the charge of the average sEPSC for each pair. Sustained exocytosis rate (in SV per s) was calculated from the slope of the linear function divided the charge of the average sEPSC. Individual recovery kinetics were determined from the ratio of probe and masker responses at 10 ms of the depolarization, with the ratio between masker and masker being 1. The recovery traces were fitted with a single exponential function from 16 to 20,000 ms to determine the time constant of RRP recovery.

Data was prepared for presentation using Adobe Illustrator. Skewness analysis and other statistical analysis were performed using GraphPad Prism 9. Statistical significance was assessed with unpaired t-test or non-parametric Mann-Whitney U test depending on the normal distribution and equality of variances of the data (Saphiro-Wilk test and F test). Data is expressed as mean ± sem unless stated otherwise.

## Acknowledgements

We would like to thank Dr. Chao-Hua Huang and Dr. Jakob Neef for their experimental and analytical input. Dr. Antoine Huet for the discussion regarding in vivo response properties of SGNs. Drs. Erwin Neher, Manfred Lindau, and Jakob Neef for critical input into this project. LMJT was a recipient of the Erwin Neher Fellowship and TM is a Max-Planck Fellow at the Max Planck Institute for Multidisciplinary Sciences. This work was further supported by the Deutsche Forschungsgemeinschaft (DFG, German Research Foundation) via the Collaborative Research Center 889 (project A02), Germany's Excellence Strategy - EXC 2067/1- 390729940 and the Leibniz Program (MO896/5 to TM), the European Research Council through the Advanced Grant 'DynaHear' to TM under the European Union's Horizon 2020 Research and Innovation program (grant agreement No. 101054467), and by Fondation Pour l'Audition (FPA RD-2020-10). LMJT is a member of the Hertha Sponer College from the Cluster of Excellence Multiscale Bioimaging (MBExC). Open access funding provided by Max Planck Society.

## Additional information

### Funding

| Funder | Grant reference number | Author |
| --- | --- | --- |
| Max Planck Institute for Multidisciplinary Sciences | Erwin Neher Fellowship | Lina María Jaime Tobón |
| Max Planck Institute for Multidisciplinary Sciences | Max Planck Fellow | Tobias Moser |
| Deutsche Forschungsgemeinschaft | CRC 889 Project A02 | Lina María Jaime Tobón Tobias Moser |
| Deutsche Forschungsgemeinschaft | Leibniz Program (MO896/5) | Tobias Moser |
| European Research Council | "DynaHear" grant agreement no. 101054467 | Tobias Moser |
| Deutsche Forschungsgemeinschaft | Germany's Excellence Strategy - EXC 2067/1- 390729940 | Tobias Moser |
| Fondation Pour l'Audition | FPA RD-2020-10 | Tobias Moser |

The funders had no role in study design, data collection and interpretation, or the decision to submit the work for publication. Open access funding provided by Max Planck Society.

### Author contributions

Lina María Jaime Tobón, Conceptualization, Resources, Formal analysis, Funding acquisition, Investigation, Visualization, Methodology, Writing – original draft, Project administration, Writing – review and editing; Tobias Moser, Conceptualization, Resources, Funding acquisition, Investigation, Writing – original draft, Project administration, Writing – review and editing

### Author ORCIDs

Lina María Jaime Tobón ⓘ https://orcid.org/0000-0002-6752-7750
Tobias Moser ⓘ https://orcid.org/0000-0001-7145-0533

## Ethics

The animal handling and euthanizing complied with national animal care guidelines and were announced to the local animal welfare committee of the University of Göttingen and the Max Planck Institute for Multidisciplinary Sciences, as well as to the Animal Welfare Office of the State of Lower Saxony, Germany (announcement T 37.03).

Reviewer #1 (Public review): https://doi.org/10.7554/eLife.93749.4.sa1
Reviewer #2 (Public review): https://doi.org/10.7554/eLife.93749.4.sa2
Reviewer #3 (Public review): https://doi.org/10.7554/eLife.93749.4.sa3
Author response https://doi.org/10.7554/eLife.93749.4.sa4

# Additional files

## Supplementary files

- MDAR checklist

- Supplementary file 1. Total time for spontaneous rate calculation.

## Data availability

Data and analysis created for the study are available at the Research Data Repository of the Göttingen Campus (GRO.data) with the DOI/accession number https://doi.org/10.25625/VUETWT.

The following dataset was generated:

| Author(s) | Year | Dataset title | Dataset URL | Database and Identifier |
|---|---|---|---|---|
| Jaime Tobón LM, Moser T | 2024 | Source data files for the article "Bridging the gap between presynaptic hair cell function and neural sound encoding" | https://doi.org/10.25625/VUETWT | GRO.data, 10.25625/VUETWT |

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
