## [Editor Report · eLife Assessment]

This **fundamental** study advances substantially our understanding of sound encoding at synapses between single inner hair cells of the mouse cochlea and spiral ganglion neurons. Dual patch-clamp recordings-a technical tour-de force-and careful data analysis provide **compelling** evidence that the functional heterogeneity of these synapses contributes to the diversity of spontaneous and sound-evoked firing by the neurons. The work will be of broad interest to scientists in the field of auditory neuroscience.

---

## [Referee Report · Reviewer #1 (Public review)]

Summary:

Tobón and Moser reveal a remarkable amount of presynaptic diversity in the fundamental Ca dependent exocytosis of synaptic vesicles at the afferent fiber bouton synapse onto the pilar or mediolar sides of single inner hair cells of mice. These are landmark findings with profound implications for understanding acoustic signal encoding and presynaptic mechanisms of synaptic diversity at inner hair cell ribbon synapses. The paper will have an immediate and long-lasting impact in the field of auditory neuroscience.

Main findings: (1) Synaptic delays and jitter of masker responses are significantly shorter (synaptic delay: 1.19 ms) at high SR fibers (pilar) than at low SR fibers (mediolar; 2.57 ms). (2) Masked evoked EPSC are significantly larger in high SR than in low SR. (3) Quantal content and RRP size are 14 vesicles in both high and low SR fibers. (4) Depression is faster in high SR synapses suggesting they have a higher release probability and tighter Ca nanodomain coupling to docked vesicles. (5) Recovery of master-EPSCs from depletion is similar for high and low SR synapses, although there is a slightly faster rate for low SR synapses that have bigger synaptic ribbons, which is very interesting. (6) High SR synapses had larger and more compact (monophasic) sEPSCs, well suited to trigger rapidly and faithfully spikes. (7) High SR synapses exhibit lower voltage (~sound pressure in vivo) dependent thresholds of exocytosis.

Great care was taken to use physiological external pH buffers and physiological external Ca concentrations. Paired recordings were also performed at higher temperatures with IHCs at physiological resting membrane potentials and in more mature animals than previously done for paired recordings. This is extremely challenging because it becomes increasingly difficult to visualize bouton terminals when myelination becomes more prominent in the cochlear afferents. In addition, perforated patch recordings were used in the IHC to preserve its intracellular milieu intact and thus extend the viability of the IHCs. The experiments are tour-de-force and reveal several novel aspects of IHC ribbon synapses. The data set is rich and extensive. The analysis is detailed and compelling.

---

## [Referee Report · Reviewer #2 (Public review)]

Summary:

The study by Jaime-Tobon & Moser is a truly major effort to bridge the gap between classical observations on how auditory neurons respond to sounds and the synaptic basis of these phenomena. The so-called spiral ganglion neurons (SGNs) are the primary auditory neurons connecting the brain with hair cells in the cochlea. They all respond to sounds increasing their firing rates, but also present multiple heterogeneities. For instance, some present a low threshold to sound intensity, whereas others have high threshold. This property inversely correlates with the spontaneous rate, i.e., the rate at which each neuron fires in the absence of any acoustic input. These characteristics, along with others, have been studied by many reports over years. However, the mechanisms that allow the hair cells-SGN synapses to drive these behaviors are not fully understood.

The level of experimental complexity described in this manuscript is unparalleled, producing data that is hardly found elsewhere. The authors provide strong proof for heterogeneity in transmitter release thresholds at individual synapses and they do so in an extremely complex experimental settings. In addition, the authors found other specific differences such as in synaptic latency and max EPSCs. A reasonable effort is put in bridging these observations with those extensively reported in in vivo SGNs recordings. Similarities are many and differences are not particularly worrying as experimental conditions cannot be perfectly matched, despite the authors' efforts in minimizing them.

---

## [Referee Report · Reviewer #3 (Public review)]

Summary:

The manuscript by Jaime Tobon and Moser uses patch-clamp electrophysiology in cochlear preparations to probe the pre- and post-synaptic specializations that give rise to diverse activity of spiral ganglion afferent neurons (SGN). The experiments are quite an achievement! They use paired recordings from pre-synaptic cochlear inner hair cells (IHC) that allow precise control of voltage and therefore calcium influx, with post-synaptic recordings from type I SGN boutons directly opposed to the IHC for both presynaptic control of membrane voltage and post-synaptic measurement of synaptic function with great temporal resolution.

Any of these techniques by themselves are challenging, but the authors do them in pairs, at physiological temperatures, and in hearing animals, all of which combined make these experiments a real tour de force. The data is carefully analyzed and presented, and the results are convincing. In particular, the authors demonstrate that post-synaptic features that contribute to the spontaneous rate (SR) of predominantly monophasic post-synaptic currents (PSCs), shorter EPSC latency, and higher PSC rates are directly paired with pre-synaptic features such as a lower IHC voltage activation and tighter calcium channel coupling for release to give a higher probability of release and subsequent increase in synaptic depression. Importantly, IHCs paired with Low and High SR afferent fibers had the same total calcium currents, indicating that the same IHC can connect to both low and high SR fibers. These fibers also followed expected organizational patterns, with high SR fibers primarily contacting the pillar IHC face and low SR fibers primarily contacting the modiolar face. The authors also use in vivo-like stimulation paradigms to show different RRP and release dynamics that are similar to results from SGN in vivo recordings. Overall, this work systematically examines many features giving rise to specializations and diversity of SGN neurons.

---

## [Author Response]

The following is the authors’ response to the previous reviews.

**Reviewer #2 (Recommendations for the authors):**
Discussion, page 28. The argument that the authors put forward justifying the (small) size of the spontaneous EPSCs seems reasonable. Nonetheless, it would be good to have an amplitude distribution constructed with voltage-evoked EPSCs to compare with that of spontaneous EPSCs. Not the large initial EPSC, obtained upon IHC depolarization but rather EPSCs occurring later during the longer pulses (figure 4). The authors made the claim that upon IHC depolarization, EPSCs sizes increased, but this is not backed with data.

Following the reviewer recommendation, we have analyzed the voltage-evoked EPSCs occurring during the last 20 ms of the Masker stimulus. We compared the cumulative distribution of the amplitude of these eEPSCs to the cumulative distribution of the amplitude of the sEPSCs (Figure 1-figure supplement 1, panel G) from the same synapses. The two distributions are significantly different (*p* < 0.0001, Kolmogorov-Smirnov test), with evoked EPSCs having larger amplitudes (average sEPSC amplitude of -97.28 ± 2.22 pA [median 82.10 pA] vs average eEPSC amplitude of 135.8 ± 3.24 pA [median 120.0 pA]).